# Identification of Sex-Specific Markers and Candidate Genes Using WGS Sequencing Reveals a ZW-Type Sex-Determination System in the Chinese Soft-Shell Turtle (*Pelodiscus sinensis*)

**DOI:** 10.3390/ijms25020819

**Published:** 2024-01-09

**Authors:** Junxian Zhu, Yongchang Wang, Chen Chen, Liqin Ji, Xiaoyou Hong, Xiaoli Liu, Haigang Chen, Chengqing Wei, Xinping Zhu, Wei Li

**Affiliations:** 1College of Fisheries and Life Science, Shanghai Ocean University, Shanghai 201306, China; zhujunxian_1994@163.com; 2Key Laboratory of Tropical and Subtropical Fishery Resources Application and Cultivation, Ministry of Agriculture and Rural Affairs, Pearl River Fisheries Research Institute, Chinese Academy of Fishery Sciences, Guangzhou 510380, China; ayongchanga@163.com (Y.W.); chenchen@prfri.ac.cn (C.C.); jiliqin@prfri.ac.cn (L.J.); hxy@prfri.ac.cn (X.H.); liuxl@prfri.ac.cn (X.L.); zjchenhaigang@prfri.ac.cn (H.C.); zjweichengqing@prfri.ac.cn (C.W.)

**Keywords:** *Pelodiscus sinensis*, whole-genome sequencing, sex-specific sequence screening, sex-specific markers, sex-determining candidate genes

## Abstract

Male and female Chinese soft-shelled turtles (*Pelodiscus sinensis*) have sex-dimorphic growth patterns, and males have higher commercial value because of their larger size and thicker calipash. Thus, developing sex-specific markers is beneficial to studies on all-male breeding in *P. sinensis*. Here, we developed an accurate and efficient workflow for the screening of sex-specific sequences with ZW or XY sex determination systems. Based on this workflow, female and male *P. sinensis* reference genomes of 2.23 Gb and 2.26 Gb were obtained using de novo assembly. After aligning and filtering, 4.01 Mb female-specific sequences were finally identified. Subsequently, the seven developed sex-specific primer pairs were 100% accurate in preliminary, population, and embryonic validation. The presence and absence of bands for the primers of P44, P45, P66, P67, P68, and P69, as well as two and one bands for the PB1 primer, indicate that the embryos are genetically female and male, respectively. NR and functional annotations identified several sex-determining candidate genes and related pathways, including *Ran*, *Eif4et*, and *Crkl* genes, and the insulin signaling pathway and the cAMP signaling pathway, respectively. Collectively, our results reveal that a ZW-type sex-determination system is present in *P. sinensis* and provide novel insights for the screening of sex-specific markers, sex-control breeding, and the studies of the sex determination mechanism of *P. sinensis*.

## 1. Introduction

In aquaculture, the Chinese soft-shell turtle (*Pelodiscus sinensis*) is the top aquatic economic species of amphibians and reptiles in China, with an annual output exceeding 300,000 tons for ten consecutive years [1,2]. However, there is marked sexual dimorphism in body growth and calipash when comparing male and female *P. sinensis*, and males have higher economic and industrial value than females [3,4,5]. Therefore, there is an urgent need to breed monosexual populations of male *P. sinensis* to benefit the entire aquaculture industry [6,7]. Currently, sex-controlled breeding has been widely applied in aquaculture species [8,9]. Nevertheless, the all-male breeding of *P. sinensis* has not seen a substantial breakthrough [10], mainly owing to insufficient studies on its sex determination mechanism.

In the last decade, the process of cognition regarding the sex-determination mechanism of *P. sinensis* has long been controversial. Previous studies have shown that the sex of *P. sinensis* depended on the post-fertilization incubation temperature and fell into the category of temperature-dependent sex determination (TSD) rather than genetic sex determination (GSD) [11,12,13]. However, more evidence has shown that *P. sinensis* experiences GSD with a ZZ/ZW type of sex chromosome [14,15,16,17,18]. Other researchers have ruled out TSD in *P. sinensis* via molecular cytogenetics and incubation experiments [19]. Therefore, further and deeper studies will contribute to clarifying the sex-determination mechanisms of *P. sinensis*.

Sex-specific genes are located on the heterogametic sex chromosomes, and they are prerequisites for revealing the sex determination mechanisms, identifying sex-specific markers, and actualizing sex-controlled breeding [20,21,22]. In traditional methods, restriction fragment length polymorphism (RFLP) [23,24], the random amplification of polymorphic DNA (RAPD) [25,26], and amplified fragment length polymorphism (AFLP) [27,28] are frequently used to develop sex-specific DNA molecular markers. However, they are gradually being phased out of the historical stage due to their complicated experimental workflow and expensive costs [29,30]. With the development of molecular biotechnology, it has become increasingly popular to develop sex-specific marker methods based on next-generation sequencing, including restriction site-associated DNA sequencing (RAD-seq) [31,32], double-digest restriction-site associated DNA sequencing (ddRAD-seq) [33,34], and type IIB endonucleases restriction-site associated DNA sequencing (2bRAD-seq) [35,36]. Meanwhile, whole-genome sequencing (WGS-seq) has become available for identifying sex-specific markers with the decreasing cost of sequencing [37,38]. Nevertheless, the development of sex-specific markers often faces problems with pseudo sex-specific markers [39]. Additionally, multiple sex-determination systems co-exist in the animal kingdom, such as heterogametic male (XY), heterogametic female (ZW), and multiple sex systems (X1X1X2X2/X1X2Y) [40], undoubtedly raising the difficulty in screening sex-specific sequences. Accordingly, it is extremely necessary to establish a scientific method and process for efficient sex-specific sequence detection.

In this study, we have developed an accurate and efficient workflow for screening sex-specific markers for the ZW or XY sex-determination system. Subsequently, based on whole-genome sequencing data from three females, three males, and a mixed male pool, genetic difference sequences between the male and female genomes of *P. sinensis* were found using this workflow and were applied to exploit sex-specific markers. Meanwhile, a simple PCR genotyping method was established using these sex-specific markers to accurately verify the genetic sex of *P. sinensis* embryos and adults. Finally, some sex-determining candidate genes in female-specific sequences were identified and functionally annotated. Our findings provide a novel insight into the screening of sex-specific sequences and will contribute to studies on sex-controlled breeding and sex-determination mechanisms of *P. sinensis*.

## 2. Results

### 2.1. Results of the Mathematical Simulation

Python scripts were used to randomly simulate the gene frequency (*p* and *q*, ranging from 0 to 1) and the calculated values of P_(E)_, with the different sample numbers of groups ranging from one female and one male to five females and five males. Each individual experiment was iterated one million times. The results showed that the mean value of P_(E)_ from the group of three females and three males was 0.0176%, obviously lower than that from the groups of one female and one male and two females and two males (*p* < 0.05, Figure 1A). Therefore, at least three females and three males were suggested as a group for preliminarily obtaining sex-specific markers and evaluating the sex-determination system of sequenced species. Furthermore, to strictly remove pseudo sex-specific markers, different numbers of single-sex mixed pools (mixed female pool used for the XY system and mixed male pool used for the ZW system) were applied for mathematical simulation. The value of r randomly selected the value of *p* (or *q*), and each individual test was iterated one million times using Python scripts. The results showed that when the number of mixed pools approached 15, the maximum value of P_(E-Pool)_ was still less than 10^−10^ (Figure 1B). Consequently, the single-sex mixed pool of at least 15 individuals was recommended for tightly removing pseudo sex-specific markers.

### 2.2. Genome Sequencing and Assembly

After data quality control, we obtained a total of about 121.67, 40.12, and 44.08 Gb in three female libraries and 131.00, 43.99, 48.27, and 163.99 Gb clean reads in four male libraries, respectively (Appendix A). The male and female reference genomes were assembled using deeply sequenced female I and male I libraries, respectively. Based on the k-mer analysis (k = 21), the estimated genome sizes of female I and male I were approximately 2.43 Gb and 2.38 G, respectively (Figure 2 and Appendix A). The final assembled genomes of female and male *P. sinensis* were 2.23 Gb and 2.26 Gb, respectively, of which the female I reference genome contained 2,792,723 assembled sequences, with an N50 length of 8068 bp, and the male I reference genome consisted of 2,952,530 assembled sequences, with an N50 length of 7837 bp (Appendix A). Three individual female and male libraries were separately aligned to the female I and male I reference genomes, with all map ratios greater than 91%, indicating a better assembly of the reference genomes (Appendix A). 

### 2.3. Screening the Sex-Specific Sequences

After the first round of mapping, a total of 5.12 Mb and 0.26 Mb of female- and male-specific sequences were obtained, which contained 1735 sequences with an average length of 2951 bp and 151 sequences with an average length of 1733 bp, respectively (Appendix A). The total length of the female-specific sequences was ~20 times longer than that of the male-specific sequences, and female-specific markers only amplified the target bands in females in the subsequent PCR validation, while male-specific markers amplified the bands in both males and females (Appendix A), initially indicating the sex-determination system of *P. sinensis* to be the ZW type. Finally, we gained 4.01 Mb of female-specific sequences in a male mixed-pool alignment, containing 1883 sequences with an average length of 2131 bp, and no male-specific sequences were obtained (Figure 3), indicating that *P. sinensis* has a female heterogametic ZW-type sex-determination system.

### 2.4. Validation of Female-Specific Markers in Adult P. sinensis

In the initial screening, seven pairs of female-specific primers (P44, P45, P66, P67, P68, P69, and PB1) successfully identified the genetic sex of 12 females and 12 males with 100% accuracy using a PCR/gel electrophoresis assay (Figure 4A and Table 1). All primers specifically amplified only one band in female individuals, except for PB1, which amplified two bands in females and one band in males. Subsequently, a total of 162 *P. sinensis* samples were collected from eight different widespread geographical populations across China and used to confirm the authenticity and universality of those seven female-specific markers. Not surprisingly, the authenticity of seven female-specific markers in these genetic sex tests was 100% (Figure 4B–I), which strongly supported the high reliability of these sex-specific markers and the true validity of these female-specific sequences. 

### 2.5. Validation of Female-Specific Markers during the Embryonic Stages of P. sinensis

In addition, two pairs of female-specific primers (P44 and PB1) were randomly chosen to confirm the genetic sex of *P. sinensis* embryos. First, the embryonic gonad-mesonephric complexes of males and females, tested by the female-specific primers P44 and PB1 (Figure 5Q–X), were randomly isolated from stage 17 (Figure 5A,B), stage 20 (Figure 5C,D), stage 23 (Figure 5E,F), and stage 26 (Figure 5G,H), respectively. Afterwards, the sex of the *P. sinensis* embryos of stage 17 (Figure 5I,J), stage 20 (Figure 5K,L), stage 23 (Figure 5M,N), and stage 26 (Figure 5O,P) was identified via histological observation. The results showed that the gonads of individuals in stage 20, stage 23, and stage 26 were morphologically differentiated and exhibited significant diversity between males and females (Figure 5K–P). Briefly, the gonads of males had a dense medulla with seminiferous cords (Figure 5L,N,P). Differently, the gonads of females had a highly developed outer cortex and a vacuolated medulla (Figure 5K,M,O). However, the gonads of individuals in stage 17 did not have obvious morphological differences (Figure 5I,J), meaning that the gonads had not yet begun to differentiate or had just started. Collectively, the female-specific primers of P44 and PB1 had a 100% accuracy when used to estimate the genetic sex of *P. sinensis* embryos in stage 17 (Figure 5Q,R), stage 20 (Figure 5S,T), stage 23 (Figure 5U,V), and stage 26 (Figure 5W,X). This is the first complete experimental evidence of using sex-specific markers to distinguish the genetic sex of *P. sinensis* embryos at different developmental stages.

### 2.6. Functional Annotation of Sex-Determining Candidate Genes

A total of 42 protein-coding genes have been annotated in the NR database (Appendix A), including *Ran*, *Eif4et*, *Crkl*, etc. The sex-determining candidate genes are significantly enriched in the catalytic activity of the molecular function, the cellular process of the biological process, and the cellular anatomical entity of the cellular component (Appendix A). Among them, 42 of the GO terms were significantly enriched, and the top three significantly enriched GO terms were DNA integration, intramolecular transferase activity, and methylmalonyl CoA mutase activity (Figure 6A and Appendix A). Meanwhile, eight KEGG pathways were significantly enriched, including the mRNA surveillance pathway, the insulin signaling pathway, adrenergic signaling in cardiomyocytes, the cGMP-PKG signaling pathway, focal adhesion, the cAMP signaling pathway, the regulation of the actin cytoskeleton, and ribosome biogenesis in eukaryotes (Figure 6B and Appendix A).

## 3. Discussion

Sexual dimorphism commonly presents in economically important aquatic species, and the identification of sex-specific markers will benefit genetic sex evaluation and sex-determination candidate gene exploration, thereby accelerating sex-controlled breeding studies [8,9]. Our novel method, based on WGS-seq, was developed, optimized, and specifically suited for the sex-specific sequence screening of diploid species with GSD. Furthermore, it has been successfully applied to developing sex-specific markers for *Channa maculate* [41], *Siniperca scherzeri*, and *Siniperca kneri* [39]. Compared to traditional sex-specific marker identification methods such as RFLP, RAPD, and AFLP, WGS-seq had a higher efficiency and broader genome coverage [42,43]. Moreover, our method, combined with mixed-pool sequencing [44,45], can use a smaller sequencing sample to obtain sex-specific sequences without any false-positive coverage, which has greatly reduced the difficulty of sample collection—especially for some cherished species [46,47]—the cost of sequencing, the time for data analysis, and the error rate of PCR validation. In a whole-genome resequencing (WGRS-seq) study of *Gadus morhua*, a total of 227 samples were used with sequencing at a depth of ~10.9 × [48]. Another disadvantage of WGRS-seq is the dependency on the reference genome of the sequenced species [49,50]; meanwhile, reduced-representation genome sequencing (RRGS-seq), including RAD-seq, ddRAD-seq, and 2bRAD-seq, shares the same shortcoming [21,51,52]. Additionally, the sex-specific sequences obtained from RRGS-seq are too few and too short [44], which is attributed to the restriction of enzyme sites, resulting in the difficulties in their detection using simple methods (PCR amplification and electrophoresis) and the risk of loss of more sex-specific sequences. For species with small differences in sex chromosomes [53], this undoubtedly greatly increases the difficulties in screening for sex-specific markers. Consequently, it highlights the potential value of our method in the species without reference genomes and fewer sex-specific sequences. Taken together, our method is an extremely powerful approach to identifying the sex-specific sequences, no matter the sex-determination system (ZZ/ZW or XX/XY) or whether the species has reference genomes.

Based on the above methods, our assembled female and male genomes were similar in size to the previously reported *P. sinensis* genome [54]. The identification of female-specific sequences and the validation of PCR results showed that *P. sinensis* is a diploid species with a ZW-type sex-determination system, which was also in agreement with previous studies [7,14,54]. Nevertheless, the sex identification of turtles has been difficult in the past, especially during embryonic development [55,56]. Some researchers have used molecular cytogenetic techniques, such as comparative genomic hybridization [57], chromosome mapping of the chicken Z-linked genes [58], and chromosome mapping of the 18S-28S ribosomal RNA genes [14], to morphologically identify the sex chromosomes and thus to evaluate the genetic sex. Moreover, a study based on quantitative real-time PCR (qRT-PCR) and a complex analytic algorithm found that the 18S gene could be used as a marker for sex identification in soft-shell turtles, with a correct classification accuracy of 85% to 90% [16]. Our results provide a powerful molecular tool for identifying genetic sex in *P. sinensis* and the first experimental evidence for using these molecular markers to identify the sex of adult *P. sinensis* and embryos.

Concurrently, we also identified 42 candidate genes for sex determination in female-specific sequences of *P. sinensis*. Although some of the star genes in sex determination such as *Dmrt1* [55], *Foxl2* [59], and *Amh* [60] were not included in that pool/group, three of them, *Ran*, *Eif4et*, and *Crkl*, gained our attention. Previous studies have identified *Ran* as a candidate gene involved in sex differentiation in *P. sinensis* [4]. *Ran* is a coactivator of the androgen receptor [61,62], and the male sex-determining gene *Sry* requires the Ran-dependent pathway to enter the nucleus [63,64]. *Eif4et* is a nucleocytoplasmic shuttling protein that interacts with *Eif4e* [65], which regulates the sex-specific expressions of *Sxl*, a master switch gene for sex determination in *Drosophila melanogaster* [66,67]. *Crkl* expression is significantly higher in ovarian cancer tissues than in normal ovarian tissues [68], and the overexpression of *Crkl* could inhibit granulosa cell differentiation and progesterone synthesis in mouse ovaries [69].

Additionally, functional enrichment analyses revealed that some candidate genes might be involved in sex-determination-related pathways, such as the insulin signaling pathway and the cAMP signaling pathway. Insulin-like androgen hormone (IAG) is an important regulator of male sex differentiation in crustaceans, and the knockdown of *IAG* resulted in the transcriptional downregulation of insulin receptors and the insulin-like growth factor binding protein [70]. The insulin receptor family is required for male sex determination and gonadal differentiation in mice [71,72], and constitutive ablation of the insulin signaling pathway leads to the down-regulation of hundreds of genes in the XX and XY gonadal differentiation programs [73]. Using cyclic adenosine monophosphate (cAMP) as a second messenger could activate the transduction of estrogen and androgen signaling [74]. In luteal cells, the luteinizing hormone specifically regulated progesterone signaling by activating the cAMP/PKA pathway and increased progesterone synthesis in response to gonadotropin stimulation [75]. Altogether, the above genes and pathways may play crucial roles in the sex determination of *P. sinensis*. However, further functional studies are needed to clarify this.

## 4. Materials and Methods

### 4.1. Theories and Mathematical Models

The screening of sex-specific markers usually encounters troubles with pseudo markers. For instance, as a diploid species, *P. sinensis* males with heterozygous genotypes in autosomes with an XY sex-determination system or females with heterozygous genotypes in autosomes with a ZW sex-determination system will produce autosomal molecular markers rather than only heterosomal molecular markers that are completely linked to sex. Therefore, we hope to circumvent the emergence of similar problems through scientific methods and processes. The details are as follows. First, the error rate of markers from the autosomes was defined as P_(E)_, the percentage of the autosomal gene A was P_(A)_ in a natural/wild population with *p* (0 ≤ *p* ≤ 1), and the percentage of allele a was P_(a)_ in a natural/wild population with *q* (0 ≤ *q* ≤ 1). Accordingly, in a diploid species, P_(A)_ and P_(a)_ equal 100 percent (P_(A)_ + P_(a)_ = 1); the percentage of genotype AA was *p2*, the percentage of genotype aa was *q2*, and the percentage of genotype Aa was 2*pq*. The distribution percentages of different genotypes among a natural/wild population of male and female individuals, in theory, are shown in Table 2. Consequently, the mathematical models of P_(E)_ with different genotypes (AA/aa/Aa) and numbers of samples (n) are shown in Table 3. Python scripts (Appendix A) were used in a simulation to obtain a suitable number of samples. Sex-specific molecular markers were preliminary screened, and the sex-determination system of sequenced species was initially tested using WGS-seq and polymerase chain reaction (PCR). Subsequently, to further verify the authenticity and universality of the molecular markers and to obtain sex-specific fragments on the heterosome, a mixed male pool sequencing (ZW system) or a mixed female pool sequencing (XY system) was carried out. In the mixed pool, the error rate of markers from the autosomes was defined as P_(E-Pool)_. The r^2^ value represents the probability of a pure male genotype (ZW-type) or a pure female genotype (XY-type) in a given deterministic sex-determination system. Hence, the mathematical model of P_(E-Pool)_ with different genotypes and numbers of samples in a mixed pool (N) is shown in Table 4. Python scripts (Appendix A) were used in a simulation to obtain suitable sample numbers of the mixed pool. SPSS Statistics 24 software was used for data analysis. Concretely, the normality of the variance distribution was tested using the Kolmogorov–Smirnov test, and a nonparametric test for multiple groups was performed using the Kruskal–Wallis test. The statistical significances are represented by a *p*-value < 0.05.

### 4.2. Experimental Turtles and Eggs

First, 33 males and 3 females of *P. sinensis* were collected on 10 March 2023, for WGS-seq and sex-related marker screening from the Huizhou Wealth Xing Industrial Co., Ltd. (Huizhou, China). Subsequently, another 24 adult *P. sinensis* turtles (12 males and 12 females) were used for the preliminary testing of sex-specific markers. Meanwhile, 162 adult *P. sinensis* turtles were collected between 22 May and 11 July 2023, from eight different geographic populations of China, including Weishanhu (11 males and 7 females), Nanning (11 males and 13 females), Changde (7 males and 5 females), Lianyugang, (10 males and 10 females), Tangshan (9 males and 12 females), Jingzhou, (12 males and 12 females), Huizhou (13 males and 9 females), and Huzhou (12 males and 9 females) (Appendix A), to further verify the authenticity and universality of sex-specific markers.

Moreover, the *P. sinensis* eggs were attained on 26 July 2023, collected from Jiayifang Aquaculture Co., Ltd. (Guangzhou, China), for a total of 200 *P. sinensis* eggs. The collected eggs were tested for hatchability, and the animal poles of the eggs had obvious white fertilization spots, which meant they had been fertilized and could hatch [76]. Fertilized eggs were hatched in a thermostatic incubator (FHX-400, Shaoguan, China) with the temperature set to 31 °C and the humidity maintained at 75~85% [77]. The gonad-mesonephric complexes of 20 embryos from different developmental stages, including stage 17 (gonadal differentiation normally begins from stage 17) [55,56], stage 20, stage 23, and stage 26, identified according to the established standards [78], were stripped away, respectively. The sex of adults and embryos was confirmed by the anatomy of the gonads and the histology of the gonad-mesonephric complexes, respectively. The liver tissue of adults and the body tissue of embryos were preserved at −20 °C for genomic DNA isolation. Genomic DNA was extracted using the Tissue DNA Kit (OMEGA, Beijing, China) while following the manufacturer’s protocol.

### 4.3. DNA Sequencing and Genome Assembly

The quality and concentration of genomic DNA were assayed with NanoDrop 2000 (Thermo Scientific, Madison, WI, USA) and 1% gel electrophoresis. High-quality DNA of three females, three males, and one mixed male pool was used, separately, for the construction of the 500 bp paired-end genomic DNA libraries using the NEBNext^®^ Ultra™ II DNA Library Prep Kit (NEB, Beverly, MA, USA) and subsequently sequenced using the Illumina XTEN platform (Illumina, San Diego, CA, USA), with a 150 bp paired-end sequencing strategy. Raw reads, including adapters > 10% of poly-N and of low quality (>50% of the bases had quality scores ≤ 5), were removed. Female I and male I sequenced to greater than a 50-fold depth of the *P. sinensis* genome (~2.20 Gb) [54] were used for de novo assembly, whereas other libraries were sequenced at a low depth of approximately 20-fold. A mixed pool of males consisting of 30 individuals was sequenced with about 160 Gb of data. The clean reads of female I and male I were assembled to the scaffold level using SOAPDenovo2 [79] based on a multi-k-mer strategy using jellyfish software (v2.3.0) [80], with the k-mer size ranging from 23 bp to 63 bp (4 bp interval) and overlapping and paired-end relationships among clean reads. The assemblies with the largest N50 were selected as reference genomes for male and female *P. sinensis*.

### 4.4. Sex-Specific Sequence Screening

Initially, the clean reads from three female and male individuals were mapped to the female I and male I reference genomes, using BWA software (v0.7.17) via the mem paired-end alignment method [81], to find female and male common sequences, respectively. The reads without mapping to the reference genomes could be removed. Subsequently, all clean reads from three male and female libraries were also compared to the female I and male I reference genomes, respectively. The female and male common sequences not covering any reads in three male and female libraries were pre-selected as candidate female- and male-specific sequences, respectively. Simultaneously, the sex-specific sequences were randomly selected to design primers, and a polymerase chain reaction (PCR) was performed to initially verify the accuracy of the markers and to confirm the sex-determination system of the studied species. If the sex-determination system is ZW-type, the clean reads from the male mixed pool are aligned to the female I reference genome. If the sex-determination system is XY-type, the clean reads from the female mixed pool are compared to the male I reference genome. This further filters the pseudo positive sequences in the first screening round, including repetitive sequences and the sequences with a higher percentage of N bases, etc., and thus obtains the final sex-specific sequences. The schematic diagram for screening the sex-specific molecular markers is shown in Figure 7.

### 4.5. Validation of Female Sex-Specific Sequences in P. sinensis

Based on female sex-specific sequences, 30 primer pairs were designed using Primer Premier 3 software [82]. To avoid nonspecific amplification, all the primers were mapped to the reference genome using bowtie v1.2.2 [83], with 2 bp mismatches at most. The genomic DNA from another 12 males and 12 females of adult *P. sinensis* was used for the preliminary screening. The total volume of the PCR was 20 μL, which contained 10 μL 2× Premix Taq (Takara, Dalian, China), 1 μL of each primer (10 μM), 1 μL of template DNA, and water to the final volume. The PCR programs were set as follows: one cycle at 95 °C for 2 min; 30 cycles at 95 °C for 30 s, 56 °C to 62 °C for 30 s, and 72 °C for 40 s to 60 s; and one cycle of 72 °C for 10 min. The amplified products were separated using 1% agarose gel, photographed with an Alpha Innotech bioimaging system (Alpha Innotech, San Francisco, CA, USA) [84], and sequenced to verify the identity of those sequences with the female-specific scaffolds. After initial screening, the candidate female sex-specific markers were eventually used to distinguish the sex of 162 *P. sinensis* turtles from eight different populations and 20 embryos from stage 17, stage 20, stage 23, and stage 26, respectively.

### 4.6. GO and KEGG Annotation

To identify potential sex-determining candidate genes, the female-specific sequences were aligned to the Non-Redundant Protein Sequence (NR) database for functional protein annotation using blastx (2.8.1), with an *E*-value less than 10^−10^. The Gene Ontology (GO) [85] and the Kyoto Encyclopedia of Genes and Genomes (KEGG) [86] databases were utilized for the functional annotation of candidate genes, with a *p*-value < 0.05 considered significantly enriched for GO terms and KEGG pathways.

## 5. Conclusions

In summary, this study developed an accurate and efficient workflow for the sex-specific sequence screening of species with XY or ZW sex-determination systems. Based on this workflow, 4.01 Mb female-specific sequences were screened in *P. sinensis*, and seven pairs of sex-specific primers were developed with 100% accuracy in the initial validation, population validation, and embryonic period validation. Moreover, some candidate genes and pathways related to sex determination in *P. sinensis* were obtained via annotating female-specific sequences. Our findings highlight the strong potential of this workflow in sex-specific sequence screening, directly demonstrate that *P. sinensis* has a ZW-type sex-determination system, and provide beneficial molecular tools for the sex-controlled breeding of *P. sinensis*, as well as potential targets for the studies of the sex-determination mechanisms of *P. sinensis*.

## Figures and Tables

**Figure 1 ijms-25-00819-f001:**
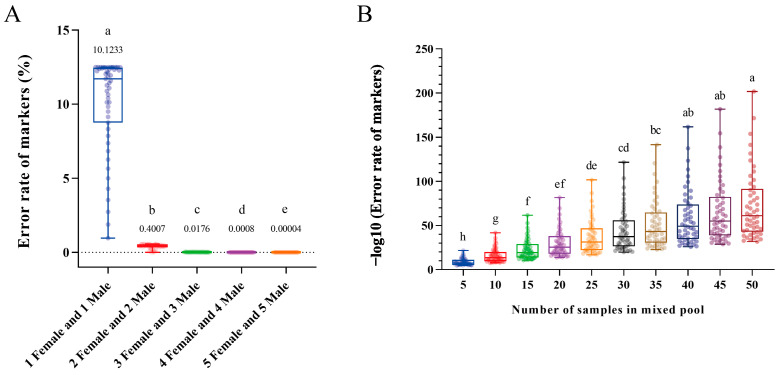
The error rates for the groups of male and female (**A**) and mixed pools (**B**) with different individual numbers. The lines in the boxes represent the median, the number above the box indicates the mean, and the whiskers indicate the minimum and maximum. The differently colored circles show each individual test with one million iterations, and different letters indicate significant differences (*p* < 0.05) among the groups.

**Figure 2 ijms-25-00819-f002:**
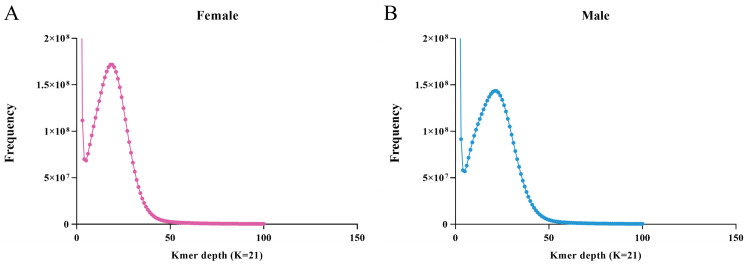
The K-mer spectrum for the genome sequences of female (**A**) and male (**B**) *P. sinensis*.

**Figure 3 ijms-25-00819-f003:**
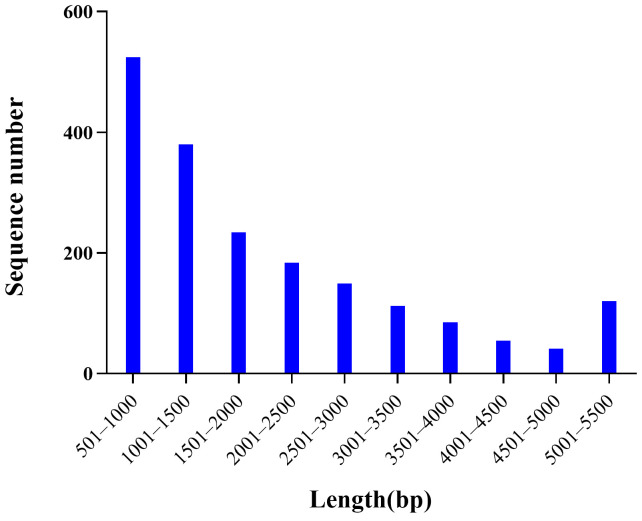
The length distribution of female-specific sequences.

**Figure 4 ijms-25-00819-f004:**
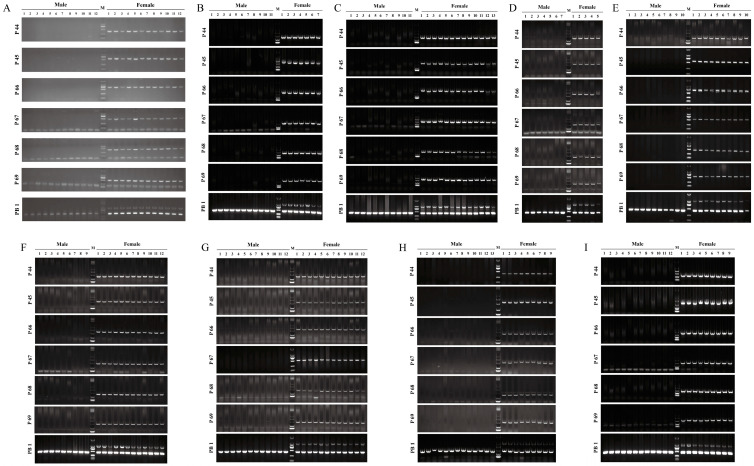
Initial (**A**) and population validation (**B**–**I**) of seven pairs of female-specific primers in adult *P. sinensis* using the PCR/gel electrophoresis assay. Plates (**B**–**I**) represent the populations from Weishanhu, Nanning, Changde, Lianyungang, Tangshan, Jingzhou, Huizhou, and Huzhou, respectively. M, DL 2000 DNA marker.

**Figure 5 ijms-25-00819-f005:**
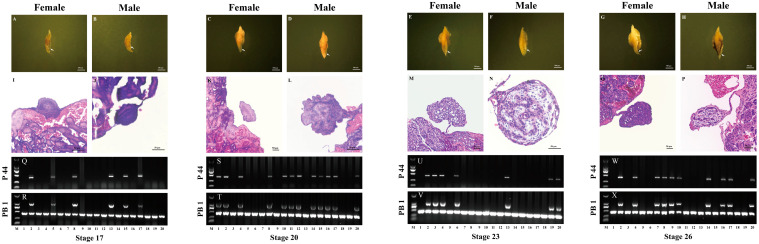
Validation of two randomly selected female-specific primers in *P. sinensis* embryos of different developmental periods. Plates (**A**–**H**) show the gonad-mesonephric complexes of *P. sinensis* embryos at different developmental stages, and the white arrows point to the gonads. Scale bar = 500 µm. (**I**–**P**) Hematoxylin and eosin staining of female and male gonads at different developmental stages. Scale bar = 50 µm. Gel electropherograms of stages 17 (**Q**,**R**), 20 (**S**,**T**), 23 (**U**,**V**), and 26 (**W**,**X**) represent the results of using sex-specific primers, P44 and PB1, to identify the genetic sex of 20 *P. sinensis* embryos in this stage, respectively. The presence and absence of bands for P44 and two and one bands for PB1 indicate that the genetic sex of the embryos is female and male, respectively. M, DL 2000 DNA marker.

**Figure 6 ijms-25-00819-f006:**
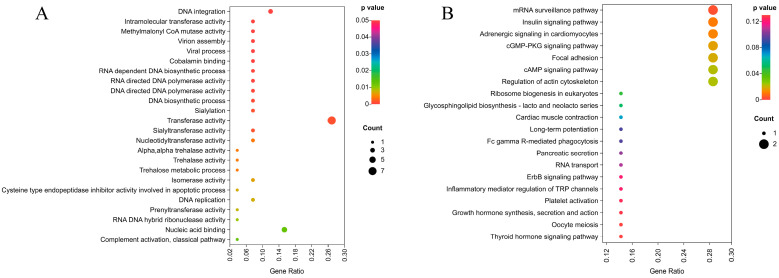
GO (**A**) and KEGG (**B**) enrichment analysis of sex-determining candidate genes.

**Figure 7 ijms-25-00819-f007:**
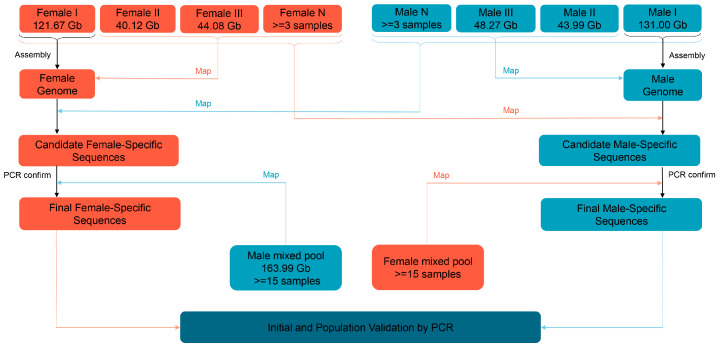
The workflow for sex-specific sequence screening in the ZW or XY sex-determination system.

**Table 1 ijms-25-00819-t001:** Candidate primer pairs for the identification of genetic sex in *P. sinensis*.

Name	Sequence (5′-3′)	TM (°C)
P44-F	GTTTCGAGTTTAGGGTTC	56
P44-R	GTGCCAATCCCTCTGTAT	
P45-F	TCTTGTAGTCGGATAGGC	58
P45-R	CAATAGACGGTTGTACTGAA	
P66-F	GCCTCGTACATTGTGATT	56
P66-R	TGTAGACAAGCCAAATCC	
P67-F	CCACCATCACCAGCACAT	60
P67-R	AAATGAGCTGGTAGTCTGG	
P68-F	CTCAAATACAGAATGGGATG	56
P68-R	CCATCCTTCGGACACTACA	
P69-F	TCCCTAAGGAGGTCTTCACG	60
P69-R	CATTCGGCTGCTTGGTGA	
PB1-F	GGATCTCATTTGTGAGCCTACATGT	62
PB1-R	CCCACAGCTTGCTTTCCWTGTTTAG	

**Table 2 ijms-25-00819-t002:** Distribution percentages of different genotypes among a natural/wild population.

GSD	Female (Aa)	Male (AA/aa)	Female (AA/aa)	Male (Aa)
ZW/XY	pq	$\frac{p^{2}}{2}/\frac{q^{2}}{2}$	$\frac{p^{2}}{2}/\frac{q^{2}}{2}$	pq

**Table 3 ijms-25-00819-t003:** Mathematical model of P_(E)_ with different genotypes and numbers of individual samples.

GSD	Mathematical Model
ZW	$P_{\left(E\right)}={\left(pq\right)}^{n}*{\left(\frac{p^{2}}{2}\right)}^{n}+{\left(pq\right)}^{n}*{\left(\frac{q^{2}}{2}\right)}^{n}$
XY	$P_{\left(E\right)}={\left(\frac{p^{2}}{2}\right)}^{n}{*(pq)}^{n}+{\left(\frac{q^{2}}{2}\right)}^{n}{*(pq)}^{n}$
ZW or XY	$P_{\left(E\right)}=2*\left({\left(pq\right)}^{n}*{\left(\frac{p^{2}}{2}\right)}^{n}+{\left(pq\right)}^{n}*{\left(\frac{q^{2}}{2}\right)}^{n}\right)$

**Table 4 ijms-25-00819-t004:** Mathematical model of P_(E-Pool)_ with different genotypes and numbers of samples in a mixed pool.

GSD	Mathematical Model
ZW	$P_{\left(E-\mathrm{Pool}\right)}=\left({\left(pq\right)}^{n}*{\left(\frac{p^{2}}{2}\right)}^{n}+{\left(pq\right)}^{n}*{\left(\frac{q^{2}}{2}\right)}^{n}\right)*r^{2N}$
XY	$P_{\left(E-\mathrm{Pool}\right)}=\left({\left(\frac{p^{2}}{2}\right)}^{n}{*(pq)}^{n}+{\left(\frac{q^{2}}{2}\right)}^{n}{*(pq)}^{n}\right)*r^{2N}$

## Data Availability

The raw data analyzed in this study can be downloaded from the National Center for Biotechnology Information (NCBI) databases (PRJNA1046242).

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
