# Peer review of "Identification of Sex-Specific Markers and Candidate Genes Using WGS Sequencing Reveals a ZW-Type Sex-Determination System in the Chinese Soft-Shell Turtle (Pelodiscus sinensis)"

_ijms, 2024, doi:10.3390/ijms25020819_

Round 1

Reviewer 1 Report

Comments and Suggestions for Authors

 Identification of sex-specific markers and candidate genes using WGS sequencing reveals a ZW-type sex determination system in the Chinese soft-shell turtle (Pelodiscus sinensis)

Rationale

Authors have developed a workflow genomic approach based on WGS-seq to investigate the sex determination system of Pelodiscus sinensis. They designed specific markers for the ZW system described and functionally annotated the sex-determining candidate genes found. They claim that the new molecular tools developed herein allow for a rapid screening of sex-specific sequences in embryos to help basic and applied sex-controlled breeding approaches.

General assessment

Sex-controlled breeding is of general concern across basic and applied disciplines for its importance in evolution, conservation and practical zootechnia. However, a large deal of controversy exists in many amphibians and reptiles regarding Genomic vs Environmental sex determination. Regardless the putative influence of temperature on sex proportions, this study comes to clarify the existence of specific genes involved in the sex determination of Pelodiscus sinensis. The English usage must be thoroughly revised. The Abstract is appalling, i.e. fully redundant and unfocused, i.e. it comprises much of Methods (the conclusions are a better abstract than this one). The abstract must be completely rewritten by briefly summarizing each section of the manuscript and stressing the main 2-3 results.

1.      Introduction

L59. Better “sex specific genes”. As author correctly say on 61-65, markers are not located but we design them on purpose. Re-write the sentence in consequence.

2. Results

L97. Remove “which”

L108 – 110. Remove the inappropriate information therein.

Fig 5. Something must be said on the nature of the samples loaded in those gels e.g. twenty embryos analyzed.

Supp. material must have the legend embedded in the illustration file and not just in the file name.

3. Discussion

L202; L216. Re-write.

L210-214. Cannot be a single sentence.

L216. Which is the “first fatal flaw?

L223-25. Re-write (What is a species with a short evol. time?

L229-37. Remove all these Methods and Results from Discussion.

L242. substitute “judge”

L243. Robert? You cannot cite by the given name!

L246. Remove “all-male breeding” for it is not a conclusion of current results.

L251. included in that pool/group...

4. Materials and Methods

In this section, please use “in a natural/wild population” instead of “among a natural population”.

In subsecton 4.2 authors must disclose the dates when all those samplings were carried out.

Strategies for sequencing, assemblage seem to be correct to me.

L295. Substitute “judged” by “tested” or “evaluated” (see also L358 ...better “pre-selected” also L100 better “evaluate”.

L297. Did you mean “wild populations”? Also, please clarify whether you used either real mixed monosex trials or simulations of mixed pools, or both.

L298-304...we carried out male mixing....” instead of....progressed.

L320-21. “...200 P. sinensis eggs...”

L362-7. Please split that overlong sentence for clarity.

L389. Better use “to identify potential...”

Comments on the Quality of English Language

The English usage in some sections is unacceptable e.g. Abstract, Discussion.

Reviewer 2 Report

Comments and Suggestions for Authors

In this study a novel methodology is developed for the sex identification in Pelodiscus sinensis turtle, a species of sexual dimorphism that affects the commercial value. The development of the method is well justified and explained in a nicely written manuscript. The fact that the method design was based on only 3 males and 3 females was overcome by the testing of the developed PCR in a number of samples for validation.

There are however some obstacles, as follows:

In the abstract the authors should explain better the designed simple PCR genotyping method that was established as a fast and easy to apply marker, in order to be easily found and sound.

I do not necessarily agree with the python calculations for the group for preliminary sex-specific markers. Also, it is not very well explained in 4.1. Particularly, the statement in lines 283-284 do not add anything and may have to be deleted. Also, in the same section, up to line 292, the authors just explain the Hardy-Weinberg equilibrium method, it should be omitted according to my opinion. In any case, since the developed PCR was tested and confirmed in a number of samples, there is no need for initial evaluation.

My major criticism is on the developed PCRs that produce band in females and no amplification in males. This could be misleading and erroneous if for some reason the PCR does not work for any reason and in this case the sample would be identified as male, but in fact could be male. Perhaps a second band with a second primer pair would verify the correct amplification or maybe a species specific band for males. As it is, the female can be confirmed but not the male. Since the purpose of the study is to present a fast and easy to apply methodology, this is important 

Table S5 presents the primers and therefore it should not be supplementary

line 341: It should probably be poly-N, not ploy-N, please correct

Reviewer 3 Report

Comments and Suggestions for Authors

Dear Authors,

the work is well presented and structured, as well as the study design that is robust and valid. The topic is a quite "niche" one, but anyway interesting for the researchers and the scientific community interested in.

My advice is to check the formulation of some sentences, also for the English better comprehension.

In Materials and Methods the statistical analysis is completely lacking, except for a simple mention in GO and KEGG annotation, such as in the Discussion section.

Best regards,

M.T

Comments on the Quality of English Language

 Moderate editing of English language is required.

Round 2

Reviewer 2 Report

Comments and Suggestions for Authors

Since the authors addressed my major point by adding the PB1 primer, and also included the rest of my suggestions in the revised manuscript, my recommendation is that the manuscript can be published in its current form